# Powerful decomposition of complex traits in a diploid model

Johan Hallin[1,*], Kaspar Märtens[2,*], Alexander I. Young[3], Martin Zackrisson[4], Francisco Salinas[1,†], Leopold Parts[2,5], Jonas Warringer[4,6] & Gianni Liti[1]

Explaining trait differences between individuals is a core and challenging aim of life sciences. Here, we introduce a powerful framework for complete decomposition of trait variation into its underlying genetic causes in diploid model organisms. We sequence and systematically pair the recombinant gametes of two intercrossed natural genomes into an array of diploid hybrids with fully assembled and phased genomes, termed Phased Outbred Lines (POLs). We demonstrate the capacity of this approach by partitioning fitness traits of 6,642 *Saccharomyces cerevisiae* POLs across many environments, achieving near complete trait heritability and precisely estimating additive (73%), dominance (10%), second (7%) and third (1.7%) order epistasis components. We map quantitative trait loci (QTLs) and find nonadditive QTLs to outnumber (3:1) additive loci, dominant contributions to heterosis to outnumber overdominant, and extensive pleiotropy. The POL framework offers the most complete decomposition of diploid traits to date and can be adapted to most model organisms.

[1] Institute for Research on Cancer and Aging, Nice (IRCAN), CNRS UMR7284, INSERM U1081, University of Nice Sophia Antipolis, 06107 Nice, France. [2] Institute of Computer Science, University of Tartu, 50090 Tartu, Estonia. [3] Wellcome Trust Centre for Human Genetics, University of Oxford, OX3 7BN Oxford, UK. [4] Department of Chemistry and Molecular Biology, Gothenburg University, 405 30 Gothenburg, Sweden. [5] Wellcome Trust Sanger Institute, Wellcome Trust Genome Campus, CB10 1SA Hinxton, UK. [6] Centre for Integrative Genetics (CIGENE), Department of Animal and Aquacultural Sciences, Norwegian University of Life Sciences, 1430 Ås, Norway. * These authors contributed equally to this work. † Present address: Millennium Nucleus for Fungal Integrative and Synthetic Biology (MN-FISB); Departamento de Genética Molecular y Microbiología, Pontificia Universidad Católica de Chile, Casilla 114-D, 8331150 Santiago, Chile. Correspondence and requests for materials should be addressed to L.P. (email: leopold.parts@sanger.ac.uk) or to J.W. (email: jonas.warringer@cmb.se) or to G.L. (email: gianni.liti@unice.fr).

Decomposing the trait variation within natural populations into its genetic components is a fundamental goal of biology that has proven to be challenging[1,2]. Environmental and gene-by-environment influences are difficult to control and alleles accounting for trait variation tend to have frequencies that are too low for their mostly weak effects to be reliably called[3]. Compounding matters, many alleles are believed to influence each other within (dominance) and between (epistasis) loci[4]. Consequently, one trait can be the result of many different allele combinations, each combination being exceedingly rare in the population. This makes the individual contributions of most alleles near impossible to assess[5]. Model organisms offer more complete dissection of complex traits because they can be analysed in controlled contexts, minimizing environmental and gene-by-environment variation, and in populations derived from a few founders, ensuring high frequencies of all alleles and allele combinations[6,7]. Because of their ease of use in genomics[8] and phenomics[9], large panels of haploid yeast segregants have allowed for fine-grained dissection of complex traits[10–12]. Unfortunately, exhaustive trait decomposition in haploid crosses requires the costly genotyping of thousands of genomes, disregards dominance and provides much simplified estimates of epistasis. A more complete partitioning of trait variation that is relevant to a diploid context has remained elusive. Inspired by previous thinking and theoretical work on recombinant inbred intercrosses in other model organisms[13–15], we here introduce a powerful and cost-effective framework for tracking the covariation through genome and phenome that allows accurate estimates of dominance and epistasis in diploid models. The framework is based on intercrossing two natural genomes over many sexual generations to reduce linkage[16,17] followed by sequencing and systematic pairing of the resulting haploid recombinant segregants to generate a very large array of diploid hybrids with fully assembled and phased genomes, termed Phased Outbred Lines (POLs). We validate the capacity of the POLs approach by genetic decomposition of growth trait variation across 6,642 diploid yeast genomes in nine distinct environments, and our results provide the most complete decomposition of diploid traits to date.

## Results

### An experimental framework for diploid complex trait analysis.
To accurately decompose diploid trait variation, we first isolated and sequenced the full genomes of 86 $MATa$ and 86 $MAT\alpha$ haploid *Saccharomyces cerevisiae* strains. These haploids were randomly drawn from a twelfth generation two-parent intercross pool, constructed using highly diverged (0.53% nucleotide difference) wild strains, here termed North American (NA) and West African (WA). Only two alleles segregate at each polymorphic site, with on average equal representation in the pool[16]. The sequenced haploids of opposite mating types were systematically crossed in all possible pairwise combinations to generate 7,396 genetically distinct diploid hybrids, retaining 6,642 POLs used for all downstream analysis (Fig. 1a, Methods).

With only a modest number of 172 haploid genomes sequenced[18], we could accurately infer the genomes of our large set of POLs. Notably, these genomes are fully phased, that is, we know the parent-of-origin for each allele and their combination into diplotypes. Furthermore, a very small fraction of genotype information is missing (max: 6.5%; mean: 0.5%; median: 0.1%; min 0%) and there are no confounding effects from segregating auxotrophies that contribute to trait variation (Supplementary Data 1). The hybrids showed remarkable uniformity, with heterozygote frequencies close to 50% (Fig. 1b). The few strong

deviations (eight deviations > 30%) from 50% heterozygosity were either due to selection for one parental allele during the intercross (overrepresentation of homozygous sites) or from the crossing design, the latter resulting in regions of fixed heterozygosity at the $MAT$ and $LYS2$ loci (Fig. 1b). Hybrid pairs sharing one haploid parent will be genetically more similar than two POLs that do not share a parent (expected fraction of loci with identical genotypes = 0.5 and 0.375, respectively), resulting in a bimodal distribution of the genetic relationship matrix entries[19].

We precisely phenotyped the complete set of 6,642 designed POLs (median CoV = 10%, mean CoV = 14%), their F12 haploid parents, the diploid NA and WA founders and their hybrid in a well replicated ($n \geq 4$) manner, using a high resolution growth phenomics platform designed to minimize noise and bias[20]. We selected nine physiologically distinct environmental conditions (Supplementary Table 1) that challenged growth to different extents (Supplementary Fig. 1a), and we obtained > 50 million population size estimates, organized into circa 250,000 growth curves (Fig. 1a, right panel). Extracting the (maximum) growth rate and a mean growth phenotype (Methods) from each growth curve (Supplementary Data 2), we found phenotype distributions across the POLs to be mostly monomodal (Fig. 1c; Supplementary Fig. 1a,b). Given the near absence of environmental variation, this implies complex traits with multiallelic influences. Growth in galactose and allantoin was bimodally distributed, in agreement with large effect sizes for the $GAL3$ (WA premature stop codon) and $DAL$ (linked loci, WA loss-of-function SNPs in $DAL1$ and $DAL4$) genes respectively[21,22]. Correlation between growth rate and mean growth ranged from $-0.13$ to 0.76 (Pearson's $r$; Fig. 1d, orange borders) but was overall low (mean $r$: 0.27; median $r$: 0.21). This agrees with the hypothesis that distinct genetic factors control population expansion in different growth phases[21,23]. Correlations across environments were positive in all but one case ($r = -0.02$) and often of moderate or large magnitude (max $r = 0.84$, median $r = 0.29$; Fig. 1d). We cannot completely exclude a small influence of shared error on correlations, but the extensive standardization, randomization and normalization (Methods), and the large variation in pairwise correlations argue compellingly in favour of extensive positive pleiotropy.

### Near complete variance decomposition of diploid traits.
Based on the *in silico* constructed diploid genomes, we used a random effects model to partition the variance in growth traits into components arising from additive (no interaction), dominance (intralocus interaction) and pairwise and third order epistatic effects (interlocus interactions) (Supplementary Note 1). We first evaluated whether the model could accurately estimate variance components as well as their uncertainty via simulation (Supplementary Note 1). The simulations showed that the model could accurately decompose the variance into additive, dominance, and pairwise epistatic components, and that s.e. estimates were well calibrated (Supplementary Data 3 and 4). When adding a component for third order interactions, the overall variance decomposition became somewhat biased, possibly due to introducing non-convexity into the optimization problem. However, the variance from third order interactions was estimated accurately (Supplementary Data 4). Due to the biasing effect, the variance decomposition for third order interactions was performed and reported separately.

The large sample size, known large variation in relatedness and absence of environmental variation allowed us to estimate nonadditive variance components with unprecedented accuracy. Thus, additivity, dominance and pairwise epistasis accounted for almost all trait variation (broad sense heritability, $H^2 = 80$–99%

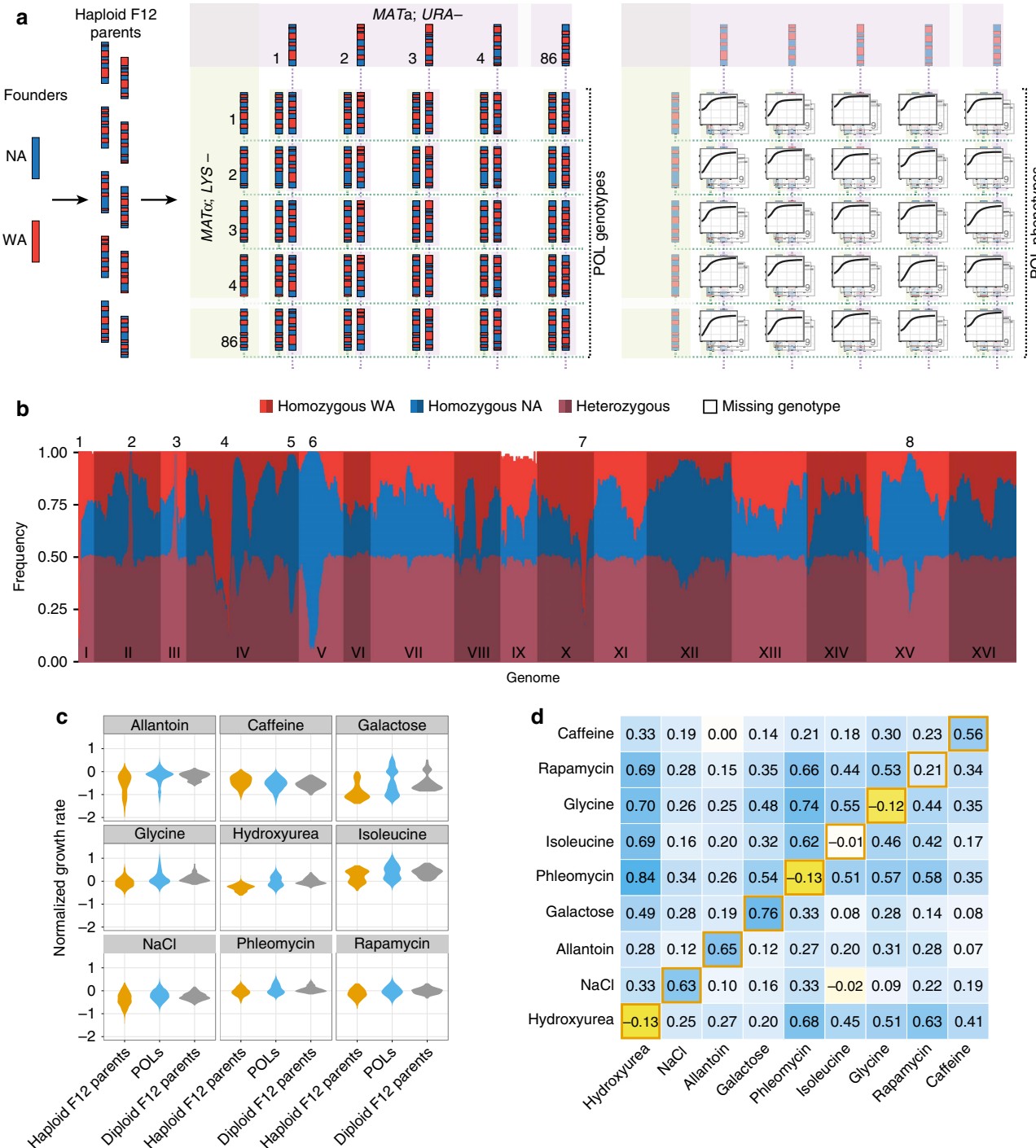

**Figure 1 | An experimental framework for analysis of diploid traits.** (**a**) Experimental design. Left panel: Advanced intercrossed lines were constructed by multiple rounds of random mating and sporulation of North American (NA) and West African (WA) genomes. Middle panel: We sequenced 172 of the resulting segregants and paired these to generate an array of 7,310 diploid hybrids (POLs). Right panel: The POLs and their F12 haploid parents were growth phenotyped in nine environments, providing high resolution growth curves. (**b**) Frequency of homozygotes (red: WA/WA, blue: NA/NA), heterozygotes (purple: NA/WA) and missing genotypes (white, mostly attributed to chr. IX aneuploidies) at each segregating site among the 7,310 POLs. Deviations from 50% heterozygosity are explained by selection (numbers 1, 4–8) against one allele in the F12 haploid parent construction, or by forced heterozygosity at the LYS2 (number 2) and MAT (number 3) loci. (**c**) Growth rate distributions of POLs (blue), their haploid F12 parents (orange) and the diploid parent estimates (grey, Methods). (**d**) Correlations (Pearson's r) between the growth rate and mean growth for POLs within environments (lower left to upper right diagonal; orange borders), between growth rates (above diagonal) and mean growth (below diagonal) in pairs of environments. Colour intensity (3-colour scale: dark yellow to white to dark blue) and number indicate the degree of correlation r.

depending on the trait, median 91%, Fig. 2, upper panel). On average, the proportion of phenotypic variance explained by additive effects was 73% (50–87%), for dominance effects this was 10% (2–45%), and for pairwise interactions this was 7% (1–15%). Complete dominance of the functional NA over the nonfunctional WA cluster of *DAL* genes[22] ensured a considerable

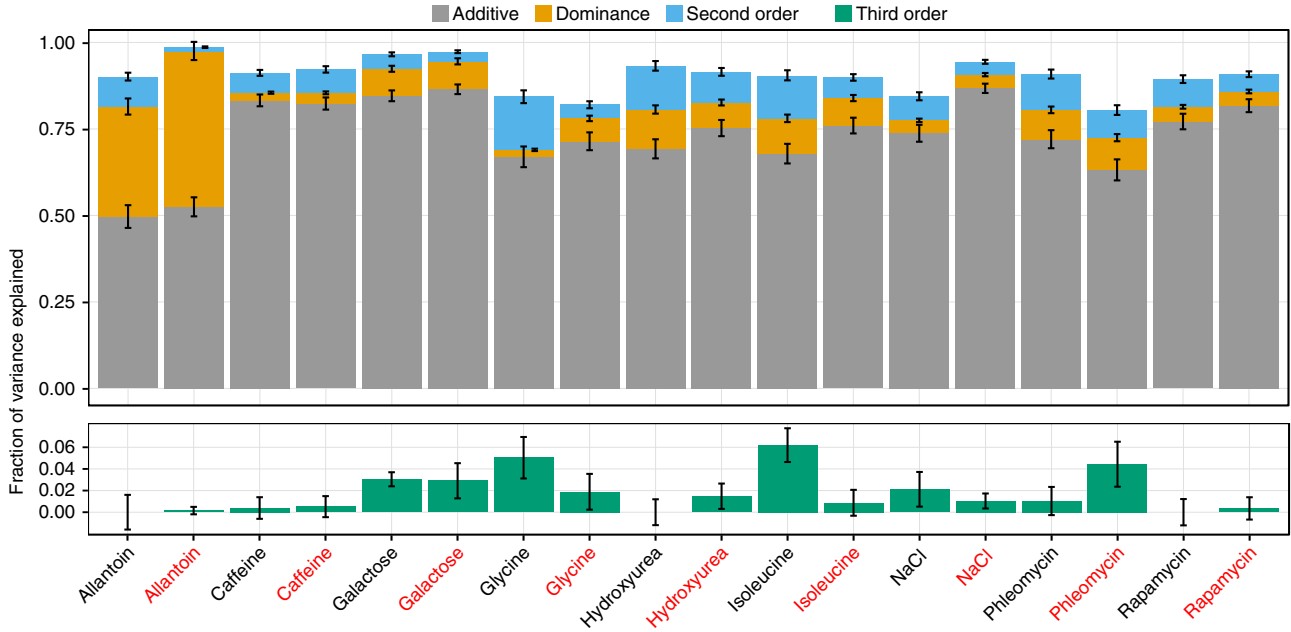

**Figure 2 | Near complete variance decomposition of diploid traits.** Decomposing the total variance in growth traits across 6,642 diploids into additive (grey, upper panel), dominance (yellow, upper panel), second order epistatic (blue, upper panel) and third order epistatic (green, lower panel) genetic contributions. Black label = growth rate, red label = mean growth. Error bars = s.e.m.

dominance component for the variation in the two allantoin phenotypes, growth rate and mean growth. Otherwise, the large variance contributions of additive genetic influences were consistent across environments (Fig. 2, upper panel).

The trait with the largest estimated variance from pairwise epistasis was growth rate on glycine (15%); this epistasis variance contribution equalled one third of the largest dominance variance estimate (45% for allantoin growth rates). We estimated that third order interactions accounted for 1.7% of the trait variation on average (Fig. 2, lower panel). However, only growth rates on isoleucine, glycine and galactose, and mean growth in the presence of phleomycin were significantly (>2 s.e.m. from 0) affected by third order epistasis. Variation in genome wide levels of homozygosity had no detectable influence on yeast fitness traits (Supplementary Fig. 2). This is in stark contrast to its substantial negative effect on human traits, for example, height[24]. Thus, the data suggest that there is no general inbreeding depression in yeast, consistent with natural populations being largely homozygous[25,26].

**Cost-efficient QTL mapping in yeast POL diploid hybrids.** Our crossing design resulted in that one haploid genome of each POL is kept constant across the 86 POLs that are derived from any one of its haploid F12 parents (Fig. 1a). This sharing of half a genome accounted for surprisingly much of the overall variation in traits, which somewhat restricted our capacity to distinguish contributions from individual alleles and allele pairs from the effect of the genetic background. Nevertheless, our platform provided a cost-efficient framework for calling both additive and nonadditive (dominance and epistasis) QTLs in diploid models. We mapped QTLs using 52,466 markers, the inferred parent phenotypes (for additive effect of genetic background) and the hybrids' deviations from the average of the inferred parental phenotypes (for nonadditive effects; Methods). Both QTL mapping approaches accounted for the population structure. We called a total of 145 unique QTLs at 10% false discovery rate (FDR) with high resolution (median 1.8-LOD support

interval = 3.67 Kbp, Supplementary Data 5). These included the *GAL3* stop codon variant, as well as the *DAL1* and *DAL4* non-synonymous and stop codon mutations, known to account for most of the variation in galactose and allantoin growth respectively (Fig. 3a, Supplementary Figs 3 and 4 and Supplementary Data 5).

Some (21%) of the QTLs contributed significantly to both additive and nonadditive phenotype components, but the majority were private to one of them (Fig. 3b). The nonadditive (75%) outnumbered the additive (25%) QTLs, but explained on average less of the variation (6 versus 28%, Student's *t*-test: $P = 2 \times 10^{-6}$, Fig. 3c, Supplementary Fig. 5). Thus, significant nonadditive trait contributions were more common but weaker. The QTLs were confirmed using linear mixed models that separated additive, dominant and epistatic effects (Methods). In almost all cases, nonadditive QTLs coincided with dominance effects (Fig. 3a). The complete recessiveness of the WA *GAL3* allele for galactose growth and of the WA *DAL* alleles for allantoin growth recapitulated established knowledge[21,22] (Supplementary Fig. 6a).

Only 32 of 145 (22%) additive and nonadditive QTLs called were mapped in a single environment, reflecting that extensive pleiotropy is the rule rather than the exception (Fig. 3d). Almost half (50 of 113, 44%) of the pleiotropic QTLs affected at least five environments, with universal growth QTLs on chr. XIII penetrating regardless of the environment and one QTL on each of chr. IX, X and XV penetrating in all but one environment (Fig. 3d). Given the wide span of environmental effects on growth and cellular physiology in our set of environments, this prevalence and penetrance of universal growth QTLs is remarkable. A surprisingly large number of QTLs (69%) were shared between growth rate and mean growth, given that the overall correlation between these growth variables was low (mean $r = 0.27$, Fig. 1d). This was to a large extent explained by the near universal chr. IX QTL affecting the two fitness components antagonistically: NA homozygotes grew slower but reached higher mean growth (Supplementary Fig. 6b). This profound fitness trade-off penetrated regardless of environment and may

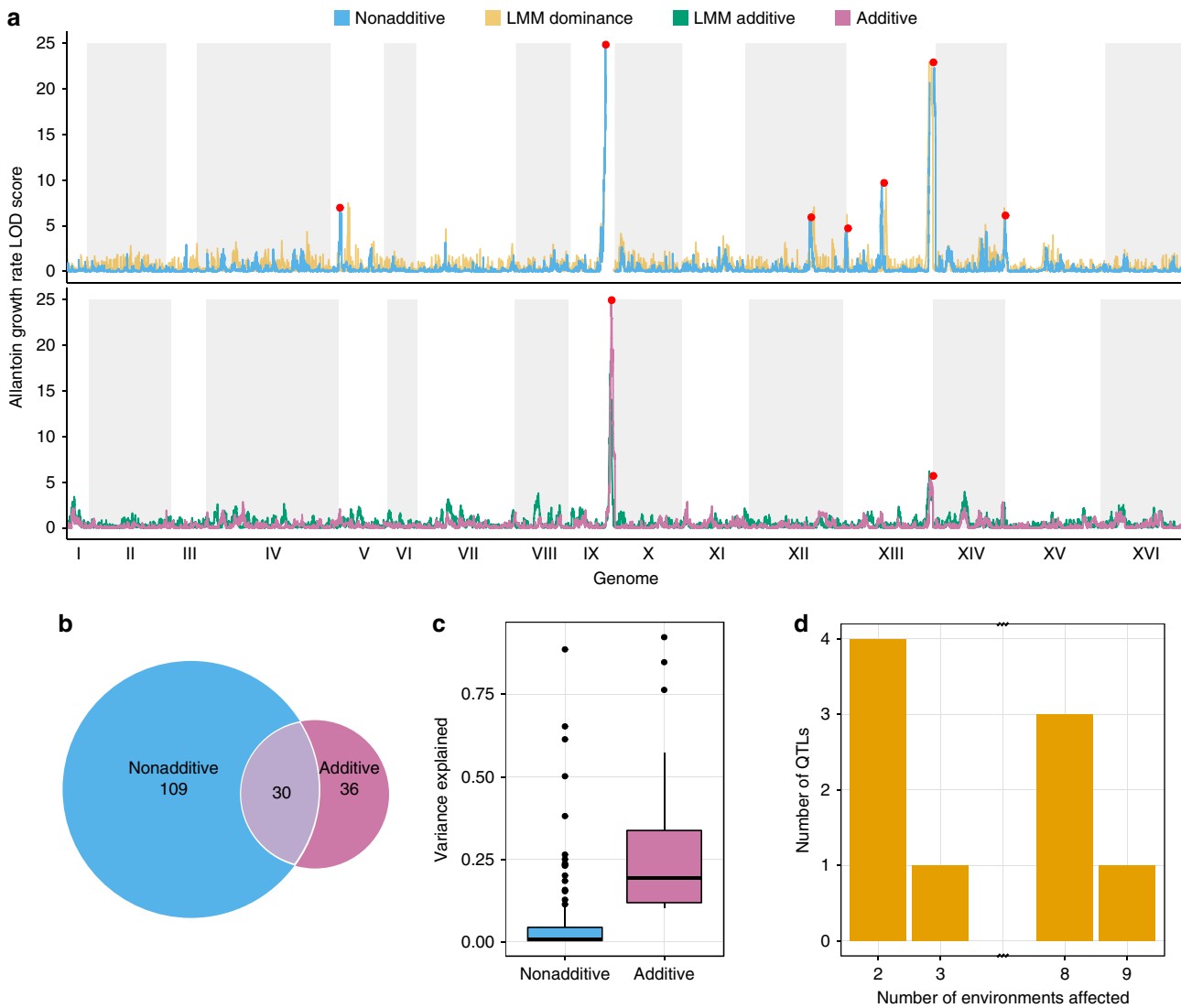

**Figure 3 | Cost-efficient QTL mapping in yeast POLs.** QTLs were mapped across 6,642 genomes and 18 traits based on additive and nonadditive contributions. QTLs were validated as additive or dominant genetic contributions using Linear Mixed Models (LMM). (**a**) QTL signal strength (LOD score, *y*-axis) as a function of genomic position (*x*-axis), for growth rate on allantoin as sole nitrogen source, using additive (LMM and non-LMM; lower panel) and nonadditive (non-LMM and LMM only capturing dominance; upper panel) models. Red dots indicate significant (FDR, q = 10%) QTL calls. White/grey fields indicate chromosome spans. (**b**) Venn diagram of significant QTLs capturing additive and nonadditive genetic contributions. All 18 phenotypes (growth rate and mean growth over nine environments) were considered, with pleiotropic QTLs counted multiple times. (**c**) Tukey boxplot showing the fraction of variance explained by additive (purple) and nonadditive (blue) significant QTLs (non-LMM models). (**d**) Histogram of pleiotropic QTLs. A QTL was counted as shared across environments if peaks were within 10 kb of each other. No QTLs were significant in 4, 5, 6 or 7 environments.

therefore have had a large influence on natural selection on the ancestral wild strains. Finally, we note that disproportionately many (28 versus 9% expected, Fisher's exact test, *P* < 0.0001) QTLs were subtelomeric; almost all (84%) of these were pleiotropic. This agrees with previous haploid studies, and adds credibility to the suggestion that hypervariable subtelomere structures and ORF compositions account for much of the remarkably large trait variation in yeast[27,28].

**Explaining heterosis by intralocus interactions.** The degree to which offspring phenotypes deviate from the mean of the parent phenotypes, heterosis, and which genetic factors that account for this difference are central questions in breeding. Capitalizing on the scale of our screen (120,000 offspring traits), we established the phenotype discordance of the POLs from those inferred for

their diploid parents (Methods) with previously unattainable completeness. Hybrid offspring where the inferred parents differed significantly from each other were retained for discordance analysis (Supplementary Fig. 7a). The majority of such offspring (89 to 95%, depending on threshold) that could be unambiguously called deviated significantly from the midparent and were thus midparent heterotic (Methods). Depending on the threshold 23–41% of these cases corresponded to the offspring being either superior (best parent heterosis) or inferior (worst parent heterosis) to both parents, with equal prevalence of best parent and worst parent heterosis (Fig. 4a). This is surprising given that earlier studies on non-recombined F1 diploids have indicated much higher prevalence of best parent heterosis than worst parent heterosis[29,30]. In these earlier studies, all recessive loss-of-function alleles are compensated for and can contribute to

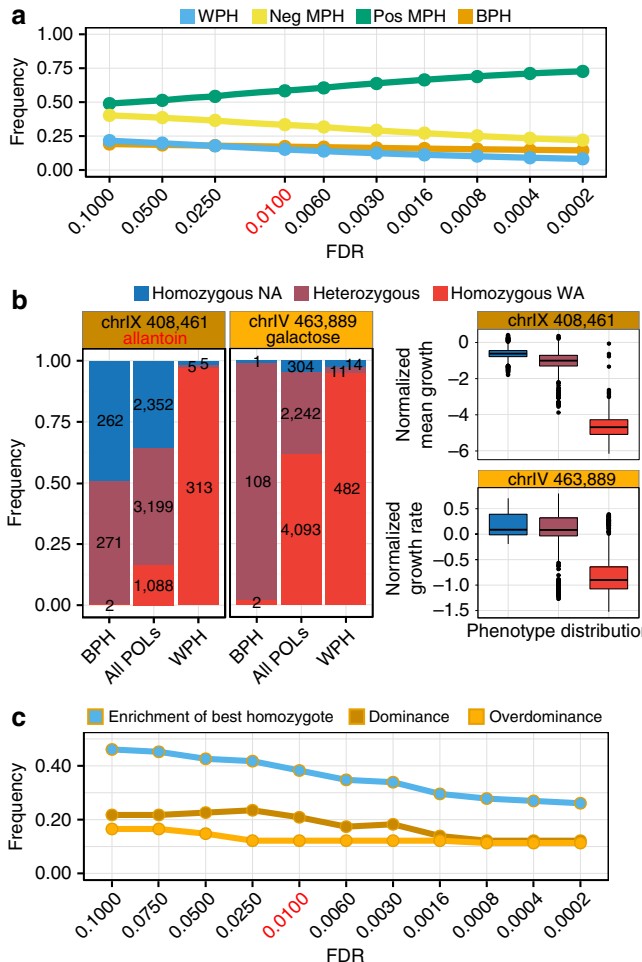

**Figure 4 | Explaining heterosis by intralocus interactions. (a)** Frequencies of the heterotic POLs (y-axis) as a function of a range of FDR significance cut-off (q) values (x-axis). Line colour = type of heterosis. Red text = FDR q-value chosen for downstream analysis (**a,c**). (**b**) Left panel: example of QTLs called as contributing to best parent heterosis by dominance (dark orange) and by overdominance (light orange) respectively. Dominance was called as enrichment of strongest homozygote and overdominance as enrichment of heterozygous state among BPH POLs as compared with all POLs (left panel). Right panel: phenotype (top: allantoin, bottom: galactose) distribution depending on genotype composition at the same QTLs. (**c**) The frequency of QTLs called as contributing by enrichment of the best homozygote, dominance and overdominance respectively (y-axis) as a function of FDR significance cutoff (q) values (x-axis). The dominance contribution is a subfraction of the contributions from enrichment of the best homozygote, dominance and overdominance respectively. Note: we show the outcomes of a range of FDR cut-off values to illustrate the robustness of conclusions; the cut-offs used for downstream analysis was set beforehand and not influenced by the results. Best parent heterosis (BPH); mid parent heterosis (MPH); worst parent heterosis (WPH).

best parent heterosis because diploid hybrids are complete heterozygotes. In our POLs, however, polymorphic sites are often homozygotic and recessive negative effects are therefore not always compensated for, explaining at least part of the difference.

Overdominance (heterozygotes at a locus being superior to both homozygotes), dominance (heterozygotes at a locus differing from the mean of the homozygotes) and epistasis can all contribute to best parent heterosis. However, calling such contributions is challenging because multiple effects often act in parallel. In particular, overdominance may be modified by

epistasis such that it only manifests in a minority of genetic backgrounds[31]. Thus, a QTL may not be overdominant in the average genetic background, but could nevertheless account for best parent heterosis in some lineages. Comparing the mean phenotypes for heterozygous and homozygous genotypes is therefore a blunt tool for detecting overdominant contributions to best parent heterosis. We devised an alternative approach, which consists of comparing the relative proportions of the genotypes among best parent heterotic POLs and the entire population of POLs. Overdominance contributions to best parent heterosis should manifest as overrepresentation of heterozygotes among best parent heterotic POLs, with no overrepresentation of either of the homozygotes. Similarly, dominance contributions should manifest as overrepresentation of the best homozygote, coupled with an unchanged or overrepresented heterozygote. Using the 115 QTLs unique to either the additive or nonadditive scan, we called overdominance contributions as more heterozygotes than expected among best parent heterotic POLs coupled with expected or less homozygotes ($\chi^2$ test, $P < 0.01$; Fig. 4b, light orange, left panel), and dominance contributions as more of the better homozygote than expected coupled with expected or more heterozygotes (Fig. 4b, dark orange, left panel). We found 44 QTLs (38%) enriched for the best homozygote genotype, and in 24 of these the heterozygote genotype was either enriched or unchanged, suggesting dominance at these 24 loci. For 14 QTLs (12%) we found overdominance contributions. These proportions were consistent across a wide range of significance cut-offs (Fig. 4c). For the remaining 50% of QTLs, no significant contributions to the best parent heterosis were detected.

The dominance/overdominance contributions of QTLs to best parent heterotic POLs were often notably different from their contributions to the population as a whole (Fig. 4b). Only two of the 14 QTLs for which we detected overdominance in the best parent heterotic POLs had, on average, a significantly superior heterozygote state when the entire POL population was considered (Student's t-test, $P < 0.01$). This suggests that dominance-by-dominance or dominance-by-additive interactions potentiate the best parent heterosis by shifting dominant or additive loci to overdominant, creating best parent heterosis, in a minority of backgrounds. For the chr. IX QTL with a near universal fitness trade-off, NA/WA heterozygotes were consistently enriched among offspring with superior growth rate, implying overdominance (Supplementary Fig. 7b). This was not the case for offspring with superior mean growth, where we instead found strong enrichment of the NA/NA homozygote, but near depletion of the NA/WA heterozygote. Finally, we called underdominant contributions to worst parent heterosis as more heterozygotes than expected among worst parent heterotic POLs. Overall, we found 7% of QTLs to contribute underdominantly to worst parent heterosis (Supplementary Fig. 7c). We also called 39% of QTLs with dominant contributions to worst parent heterosis as more of the worst homozygote state than expected coupled with an enriched or unchanged fraction of the heterozygote state. To our knowledge, this is the most exhaustive dissection of heterosis to date.

## Discussion

Traits have been exhaustively mapped and decomposed in haploid models[10–12,32,33] but extrapolation from haploid screens to the biology of diploids is precarious. Haploid designs cannot be used to measure intralocus interactions in the form of dominance, further, they only capture additive-by-additive epistasis. Moreover, ploidy has a fundamental impact on traits[34], both due to its influence on cell size and the masking of recessive

alleles in diploids[35,36]. The Phased Outbred Lines (POLs) presented here circumvent the shortcomings of haploid screens by offering decomposition of diploid traits with previously unattainable exhaustiveness. The capacity of the approach follows from generating a very large array of fully phased diploid genomes based on short read sequencing of only a moderate number of haploids. The alternative, acquiring phased genomes from direct sequencing of diploids, would require long-read sequencing of thousands of isolates and will remain economically unfeasible even in model organisms for years to come[37]. As a direct consequence of our experimental design, each POL shares one haploid genome with siblings spawned from the same haploid parent. This sharing of half a genome had surprisingly large effects on trait similarity, greatly aiding both trait prediction from relatives[19] and the partitioning of trait variation into its additive, dominant and epistatic components. In contrast, it somewhat restricted our ability to distinguish the weaker effects of individual loci and the calling of those QTLs. The large impact that sharing one haploid genome has on trait similarity among diploids, and the associated benefits and drawbacks, may or may not manifest in other model organisms. Beyond the removal of the sex-switch (HO gene) and introduction of sex-specific auxotrophic markers, POLs impose no requirements on the yeast genotypes used; the design is lineage agnostic. However, removal of the yeast sex-switch renders the cross directional and prevents the construction of a full diallel cross, something that is otherwise possible in for example monoecious plants where individuals express both sexes. The diploid hybrids have identical marker composition, avoiding growth effects derived from artificial auxotrophies that confound many haploid crossing designs[38,39].

The framework allowed partitioning diploid trait variation into its major components with little room for confounding effects, due to nearly all trait variation being accounted for. Additive effects explained the vast majority of phenotypic variation, with approximately equal variance contributions from dominance and pairwise interactions at around 10% and 7%, respectively. The large explanatory power of additive genetics is well in line with findings in haploid screens[10,33]. Third order epistasis explained <2% of the trait variation, comparable to, or somewhat less than, estimated for third[11], or third and higher[12] order interactions in haploid yeast. Thus, although examples where three-way interactions affect trait variation can be found[12,40,41], and can explain extreme phenotypic outliers[42] they generally account for little trait variation. Despite the lower overall contribution of nonadditive compared with additive genetics to trait variation, we found nonadditive QTLs to outnumber additive QTLs. The weaker mean effect of nonadditive QTLs partially explains this discrepancy. In addition, differences in how QTLs were called means that we cannot completely exclude that we detected nonadditive effects with somewhat better power.

A stable haploid phase, indefinite storage as frozen stocks and easy mating will remain distinct advantages of yeast. Nevertheless, POLs can be employed in most higher model organisms, with only slight modifications to the approach. Panels of extensively recombined offspring can be generated using two or more founder parents in mouse, plants, flies and worms[43,44]. Successive inbreeding or selfing is common practice to produce recombinant inbred lines (RILs). The gametes of these sequenced RILs can be paired by designed mating to generate the final array of POLs to be phenotyped. Somewhat analogous approaches exploiting near isogenic lines, or immortalized F2 populations, have been used in plants[45–47], although few individuals, genetic markers and recombination events and remaining segregating heterozygosity prevented both powerful decomposition of trait variation and highly resolved mapping of QTLs. Furthermore, genome phasing information in POLs derived from higher organisms is ideal for investigating parent-of-origin contributions to complex trait variation[48]. To attain exhaustiveness while avoiding confounding effects from uncontrolled environmental variation, the cost-effectiveness of the genotyping needs to be matched by a phenotyping approach that achieves both scale and accuracy. The here reached broad sense heritability, with a lower bound mean estimate of 91%, may remain challenging to match in most species. Nevertheless, phenomics is advancing on broad fronts and simultaneous high throughput and accuracy is on the horizon in most model organisms[9].

## Methods

**Generation of phased outbred lines.** F12 outbred lines were derived from a multigeneration two way intercross between ancestors of the North American (YPS128) and West African (DBVPG6044) populations, as described[16]. Ancestral strains differed at 0.53% of nucleotide sites[49]. Following random sporulation of F12 diploids, 86 stable haploids of each mating type were randomly isolated and their mating type and auxotrophies determined. Haploid genotypes were selected to allow systematic crossing: MATa, ura3::KanMX, ho::HygMX and MATα; ura3::KanMX; ho::HygMX; lys2::URA3. Haploids of different mating types were robotically mated on rich medium (1% yeast extract, 2% peptone, 2% glucose, 2% agar) in all pairwise combinations combining their complementary LYS and URA auxotrophies using a RoToR HDA robot (Singer Ltd, UK). Haploid cells of the same mating type do not mate and this feature prevents the construction of a full diallel cross (e.g., MATα/MATα and MATa/MATa diploid hybrids cannot be constructed). Diploid hybrids were selected twice on Synthetic Minimal (SM) medium (0.14% Yeast Nitrogen Base, 0.5% ammonium sulphate, 2% (w/v) glucose and pH buffered to 5.8 with 1% (w/v) succinic acid, 2% agar). The theoretical maximum amount of POLs from our experimental design was 7,396 (86 × 86); however, one F12 haploid strain (MATα, number 45) was contaminated prior to mating and all 86 hybrids spawning from this cross were therefore discarded (86 MATa × 85 MATα = 7,310 were retained). Furthermore, 8 F12 haploids were identified as having chr. IX aneuploidy (see 'Genotype construction' below), the hybrids spawning from these haploids were included in the phenotyping in order to investigate the aneuploidy's effect on the phenotype. They were, however, excluded in all downstream analysis since they could interfere with the QTL mapping and they have a large fraction of missing genotypes on chr. IX. We do find a possible effect of the chr. IX aneuploidy mainly on the mean growth phenotype (see Supplementary Fig. 1a, bottom panel).

**Genotype construction.** The haploid F12 parents were previously sequenced by short read sequencing, and mapped to the S288C reference genome in order to call segregating sites, infer genotypes and characterize the recombination landscape[18]. All segregants were homoplasmic, carrying the same non-recombined WA mtDNA genome. This excludes confounding mtDNA inheritance effects since this is inherited randomly in a yeast hybrid from only one of the two parents. Chr. IX aneuploidy was identified based on higher sequencing coverage and higher fraction of heterozygous polymorphic sites compared with the genome as described in Cubillos et al.[17]. The following eight haploid F12 parents carried the aneuploidy: MATα 41, 53, 67 and MATa 206, 222, 223, 253, 258. Contaminated diploid hybrids and hybrids with chr. IX aneuploidies were excluded. Phased genomes of the 6,642 diploid hybrid offspring (81 MATa x 82 MATα) retained for the genetic analysis was constructed in silico using custom R code.

**High resolution growth phenotyping.** High resolution growth phenotyping on solid agar medium was performed using a 1536-colony plate layout. Each plate (Plus plate, Singer Ltd, UK) was cast with exactly 50 ml of Syntetic Complete medium at 50C (as SM above with added 0.077% Complete Supplement Mixture (CSM, Formedium)). Casting was performed on an absolutely leveled surface with drying for ~1 day. The base medium was supplemented with additional stressors or alternative carbon or nitrogen sources as indicated (Supplementary Table 1). The 7,310 POLs were distributed over 1,152 positions across eight plates. We used n = 4 replicates for each experimental plate, with replicates initiated from two different pre-cultures and run in different instruments and plate positions to minimize bias. Their 172 haploid F12 parents (n = 6 replicates on each plate, two plates) and their diploid NA and WA ancestral lineages (n = 72 replicates on each plate, two plates) were phenotyped separately. Every 4th position was reserved for internal controls (diploid NA ancestral strains). These 384 controls were interleaved with experiments on pre-culture plates, ensuring equal treatment of controls and experiments. High resolution population size growth curves were obtained using Epson Perfection V700 PHOTO scanners (Epson corporation, UK) and the Scan-o-matic framework[20]. Scanners were maintained in a 30 °C, high humidity environment that minimized light influx and evaporation. Experiments were run for 72 h, with automated transmissive scanning and signal calibration in 20 min intervals. Calibrated pixel intensities were transformed into population size measures by reference to cell counts obtained by optical density measurements on

diluted samples. Raw population growth curves were slightly smoothed using a median (size = 5) and a Gaussian (width σ = 1.5) filter to remove noise. Poor quality curves (1%, descending from, for example, positions lacking colonies) were rejected following manual inspection[20]. Retained population growth curves were broken down into two growth phenotypes: (i) growth rate, extracted using linear regression from the steepest slope of the population's exponential phase, and (ii) mean growth, extracted as the area under the curve relative to its starting point but excluding the three first time points. To counter spatial bias on each 1,536 plate, the two growth phenotypes were normalized to the internal controls using the Scan-o-matic principle[20]. The final phenotypes used were the average phenotype across all replicates. Detailed protocols are available for the entire phenotype acquisition[20]. To circumvent the problem of calculating Coefficients of Variation (CoV) for normalized growth phenotypes spanning over both negative and positive values, these were reverted back into actual doubling times and yields, before CoV calculations. This reversion was performed by multiplying each normalized value with the median control trait value and reversion of the log transformation.

**Phenotype variance partitioning.** We estimated additive relatedness from genotypes. We derived formulae for efficient computation of the covariance due to dominance, pairwise and third order interaction effects (Supplementary Note 1). We fitted the model using restricted maximum likelihood, as in Yang et al.[50]. The variance decomposition and its associated standard errors were found to be accurate and close to unbiased in simulations when fitting additive, dominance, and pairwise interaction components (Supplementary Note 1). However, when adding a component for third order interactions, the overall variance decomposition became biased, even though the estimates of the third order component did not. We believe this may be the result of non-convexity in the optimization problem, as evidenced by bimodality in the distribution of estimates of pairwise interaction variance in simulations including the third order component. We therefore report estimates of the variance from third order interactions separately from the decomposition into additive, dominance and pairwise interaction components.

**QTL mapping.** QTL calling was performed using the 'scanone' function with the 'marker regression' method in R/qtl (ref. 51) with estimated diploid parent phenotypes (additive genetic background contribution to traits) and POL deviations from the estimated diploid parents values (variation not explained by additive effects of parental background) respectively using the full set of 52,466 markers (including redundant markers). Diploid parental phenotypes were estimated as the median of all hybrids that descended from that parent. Using the deviations from expected midparent phenotype for the POLs has the additional critical benefit of effectively accounting for population structure by removing the additive effect of the more similar genetic composition due to shared parents. Significance thresholds were given by permutations ( × 1,000), 1.8-LOD support intervals were calculated for each QTL using the 'lodint' function in R/qtl, this corresponds to the LOD support interval stated as the preferred one for intercrosses in A guide to QTL Mapping by Broman et al.[51]. QTL calling by linear mixed models, also accounting for population structure, was performed and used as verification. For these, in order to test each QTL, we constructed the realized genetic relationship matrix by discarding the SNPs within the 50 kb neighbourhood of the SNP under consideration; these models were fitted with LIMIX (ref. 52). Consecutive markers having the same genotype across all individuals were removed for increased computation speed, leaving 10,726 segregating sites[19]. We accounted for population structure in the LIMIX analysis by using the genetic relationship matrix defined by $K = \frac{1}{c} X X^T$ where $X$ is a centred and standardized genotype matrix, and the normalizing constant $c$ is the average diagonal value of $X X^T$. This is in contrast to the mapping in R/qtl where we instead modified the phenotype used, as stated at the beginning of this section. QQ-plots (Supplementary Fig. 8) confirm that the linear mixed models appropriately account for population structure: apart from the locus with the strongest effect (*DAL* and *GAL* loci, in allantoin and galactose respectively), the distribution of the rest of *P* values follows the expected uniform distribution under the null.

**Heterosis.** We used a Student's *t*-test to detect POLs significantly deviating (α < 0.01) from the mean parent phenotype, either overperforming (positive mid parent heterosis) or underperforming (negative mid parent heterosis). The parent phenotypes used were estimated from all POLs descending from the given parent as described under 'QTL mapping' in Methods, the variance of the mean parent phenotype was set to equal that of the most variable parent. POLs deviating from the mean parent were then tested using a Student's *t*-test (α < 0.01) for positive deviations from the strongest parent (best parent heterosis, BPH) and for negative deviations from the weakest parent (worst parent heterosis, WPH). Hybrids deviating significantly from the two parents, but not from the estimated mid-parent, was called as not deviating from the mid parent expectation. Hybrids not falling into any of the stated categories were set as ambiguous and not considered, this might manifest as for example a hybrid not being significantly different from either parent.

**Genetic contributions to heterosis.** To test for overdominance contributions to best parent heterosis we compared the expected and observed number of heterozygous genotypes among best parent heterotic POLs (defined as above). Calling overdominance as overrepresentation of the heterozygous state with no overrepresentation of either homozygous state. This was performed for each QTL separately using a $\chi^2$ test, 115 QTLs were used, corresponding to all unique QTLs between the additive and nonadditive QTL scan. Entries to the $\chi^2$ test were: observed number of heterozygotes and observed number of homozygotes (summed) among BPH POLs and the corresponding expected numbers, given distributions among all POLs. A range of cut-offs for significance was tested and the stability of results across cut-offs ascertained. We cannot completely exclude that pseudo-overdominance, that is, tightly linked loci with dominance of opposite parental alleles, confuse some assignments of overdominance. However, given the small linkage regions, we expect pseudo-overdominance to be rare and the associated overestimation of overdominance to be small. We tested for dominance similarly, but pooling the weaker homozygote state with the heterozygote state and calling significant enrichment of the better homozygote among BPH POLs. If the better homozygote was enriched, and the weaker was not, cases where the fraction of heterozygous was unchanged or enriched were called as dominance. Underdominance contributions to worst parent heterosis were called as for overdominance, but as enrichments of the heterozygous genotype among worst parent heterotic POLs. Finally, dominance contributions to worst parent heterosis were called as for dominance in best parent heterosis, but as enrichment of the weaker homozygote.

**Data availability.** All data associated with this study is available in Supplementary Information of this publication. We used R, complemented with various packages[53–58], for the analyses. The associated code can be found at https://github.com/j-hallin/y10k, and is available upon request.

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

## Acknowledgements

J.H. was supported by the Labex SIGNALIFE (ANR-11-LABX-0028-01) and K.M. was supported by the European Regional Development Fund through the BioMedIT project, F.S. was supported by ATIP-Avenir (CNRS/INSERM), Becas Chile, CONICYT/FONDECYT (3150156) and MN-FISB (NC120043) postdoctoral fellowships. This study was funded by the Swedish Research Council (325-2014-6547 and 621-2014-4605), the Research Council of Norway (222364/F20) to J.W.; by a Marie Curie International Outgoing Fellowship, the Wellcome Trust, and Estonian Research Council (IUT34-4) to L.P.; ATIP-Avenir (CNRS/INSERM), ARC (grant number PJA20151203273), FP7-PEOPLE-2012-CIG (grant number 322035), ANR (ANR-13-BSV6-0006-01 and Labex SIGNALIFE ANR-11-LABX-0028-01), Cancéropôle PACA (AAP émergence 2015) and DuPont Young Professor Award to G.L.

## Author contributions

J.H., J.W. and G.L. conceived and realised the crossing design. J.H. established the resource and generated data with help from M.Z. and F.S. J.H., K.M. analysed the data and A.I.Y. performed the variance decomposition. L.P., J.W. and G.L. supervised the project. All authors wrote and approved the manuscript.

## Additional information

**Competing financial interests:** The authors declare no competing financial interests.

**Publisher's note**: 

