## [Peer Review File · Nature Communications]

Reviewer #1 (Remarks to the Author):

This manuscript by Hallin et al is an important and impressive step forward for the field of quantitative genetics. They have taken full advantage of yeast genetics, together with high-throughput genotyping and phenotyping, to assemble an integrated view of the genetic architecture of growth rates across 9 conditions. The paper is well-written, and the figures are clear and informative.

My only significant concern is about the chr IX aneuploidy. Why were these strains only excluded from QTL analysis, and not from all analysis? How was the aneuploidy detected, and at what stage in the experiment did it appear? It seems that it wasn't as simple as chr IX being aneuploid in some subset of the 86 parents of each mating type, since in that case I'd expect the number excluded (668) to be evenly divisible by 86. Even though it's only one chromosome, it has quite a large effect on a number of traits (Fig 3a and S3), so may have a large effect on genetic architectures as well.

Minor comments:

1. "More QTLs (69%) were shared between maximum doubling time and mean growth than expected from their low general correlation (mean r : 0.27, Fig. 1d)."

How many are expected from the correlation? How is such an expectation calculated?

2. "Raw population growth curves were slightly smoothed to remove noise"

How was this done?

3. "Poor quality curves (0.3%) were rejected following manual inspection."

What constituted poor quality?

4. Fig 1 legend: "gorwth" typo

Reviewer #2 (Remarks to the Author):

In this MS an analysis is presented of a large reciprocal F1 cross between two independent sets of 86 haploid yeast strains, ie 7396 potential F1 offspring. The $2 \times 86 = 172$ parents were F12 haploid recombinant inbred lines from a two-parent cross. Thus in effect the chromosomes are all mosaics of a two-haplotype system. Growth traits were measured in replicates in these offspring under different environments. Genetic analysis to partition heritability and map QTLs was performed.

The idea of making what is in effect a large diallele cross is not new - see for example the RIX paper <http://www.ncbi.nlm.nih.gov/pubmed/26669442> for a theoretical analysis - but so far as I know this is the first study to actually collect phenotypic data and to analyse it. It would therefore be of interest to the statistical genetics readership of Nature Communications.

Judging by the large heritabilities obtained, the phenotyping seems to have been performed with great accuracy. The results as presented are focused on statistical genetics questions - estimates of heritability, heterosis etc. Although many QTLs are mapped, it's unclear if the functions of any novel genes have been confirmed.

However, I did have a number of problems with the current MS.

1. The paper is extremely brief, which makes parts of it very hard to follow. I think the page limits for Nature Communications Articles may be partially responsible but there is an over-long discussion at the cost of very opaque results (I found the discussion of heterosis in lines 187-203 particularly difficult). Similarly it was only after carefully reading the supplemental material that it became clear that the experimental design involves 86 haploids from each mating type - I initially

thought that a full diallele cross of 172 lines was being analysed.

2. Although the experimental design is presented as being generally applicable outside of the yeast genetics community, many of the details would be unclear to non-specialists (eg the constraints due to mating type). For example, if one were to repeat this design in a monoecious species such as Arabidopsis, there would be no need to distinguish the two sets. It's also worth pointing out the fact that in a diallele of diploid RILs (eg mice, Arabidopsis) the F1 genotypes are still trivial to infer from the parents.

3. The introduction of the (apparently) new terminology POL (phased outbred line) seems unnecessary and is puzzling as it over-emphasizes the phased nature of the genotypes. Although the genomes are trivially phaseable with respect to the parents, no use is made of this phasing in the MS. For example, it would be straightforward to test for parent-of-origin effects by exploring asymmetries in the phenotypes of lines carrying the phased genotype XY at a given locus vs those carrying genotype YX. Perhaps the authors did this but did not find anything? It would be interesting to know. Similarly, the long range phasing in this population means it would be possible to explore epistasis where there is a difference between a pair of loci with phased genotypes (AB - AB) compared to (AB- BA). One could presumably partition the epistatic component of heritability into these sub-components (the methodology described in the supplemental note suggests this would be possible).

In Figure 2 heritability is partitioned into additive, dominance, and 2-way epistatic components. 3-way components are plotted separately. Why was this? In the supplemental note simulations suggest that incorporating 3-way interactions distorts the estimation of the other variance components. Was this the reason? By eye it looks like the 3-way epistatic variance must be confounded with some of the other components since the total variance when 3-way epistasis is included appear to exceed 100%. In the upper panel of Figure 2, it would be helpful to add lines on the bars indicating the total variance explained by mapped QTLs.

4. As the authors point out, but in a somewhat cryptic manner, and as other have pointed out (eg in the RIX paper cited above), the problem with the POL/RIX design is that the number of independent genomes is given by the number of parents (in this case 172). This means that all pairs of F1 individuals that share a common parent have much more allele sharing than two individuals that don't. Consequently the entries of the genetic relationship matrix of the population have a bimodal distribution (trimodal if one includes the main diagonal - it would be interesting to see it as a supplemental figure). The population resembles a large population of halfsibs in which every male mates with every female. It is implicitly claimed that the mixed model used to do the QTL analysis adequately controls for the background relatedness but I would like to see some evidence. Some QQ-plots for a few genome scans would help. Similarly, does the design introduce any biases in the estimates of heritability? Is it certain that epistasis is estimated correctly, given the long-range correlations in the genotypes? I guess the simulations in the supplemental note address this, but this should be discussed in the main text.

5. As noted above, the discussion of heterosis is hard to follow. I wonder if would be better to simplify it and not subdivide it into over and under-dominance. Some way of displaying the variance attributable to heterosis in Figure 2 would be helpful. In this study, the phenotypic value of a given parent is estimated by taking the mean of all F1s with that parent. This definition is sensible from the point of view of establishing genome-wide levels of heterosis, but if one is investigating heterosis at a particular QTL then perhaps it is better to estimate the expected phenotypic value of AA genotypes vs BB genotypes (where A is the allele carried by the NA strain and B that by the WA strain), from the means of all F1s that are AA vs BB. Would this definition have resulted in substantially different conclusions regarding heterosis?

6. Was the set of SNPs tested in the GWAS - 56k in the R/qrtl analysis, followed by only 10k in LIMIX after pruning - the total number segregating in the population? What was the advantage of

using 56k SNPS in the first stage is the pruning removed SNPs perfectly tagged (in the LD sense) with others? Were there more SNPs detected during sequencing? If so, what was done with them? the numbers of SNPs tested should be mentioned in the main text.

7. More details of the R/qtl mixed model used for QTL mapping should be given. What was the definition of the kinship matrix? Was this the same as that used in LIMIX? The choice of the LOD drop threshold used to define confidence intervals should be justified.

Response to reviewer comments

Reviewer comment

Author response

Revised manuscript text

Reviewer #1

This manuscript by Hallin et al is an important and impressive step forward for the field of quantitative genetics. They have taken full advantage of yeast genetics, together with high-throughput genotyping and phenotyping, to assemble an integrated view of the genetic architecture of growth rates across 9 conditions. The paper is well-written, and the figures are clear and informative.

My only significant concern is about the chr IX aneuploidy. Why was these strains only excluded from QTL analysis, and not from all analysis? How was the aneuploidy detected, and at what stage in the experiment did it appear? It seems that it wasn't as simple as chr IX being aneuploid in some subset of the 86 parents of each mating type, since in that case I'd expect the number excluded (668) to be evenly divisible by 86. Even though it's only one chromosome, it has quite a large effect on a number of traits (Fig 3a and S3), so may have a large effect on genetic architectures as well.

Author response: We agree with the reviewer that this aspect was unclear in our original submission and after reconsideration we decided to exclude the hybrids carrying aneuploidies from all analyses and to remove all mention of it from the main text. Instead, we have added all details concerning sample exclusion to the Methods section (page 6-7, lines 298-306; page 7, lines 312-317), explaining how the aneuploidies were detected and why they were excluded. In this way, there is no ambiguity in the number of hybrids used for the analyses. We also disclose which parents that carry the aneuploidy and we are more clear on the final number of hybrids included in the downstream analysis. We have added a new panel to Supplementary Figure 1a to show the effect of aneuploidy on the phenotype distributions.

We reanalysed variance decomposition, heterosis and dominance, excluding the aneuploid hybrids (consistently with the QTL mapping). Figures and values have been changed to accordingly (Fig 2, 4, Supplementary Fig S1, S6). All conclusions remained unaltered. The large QTL contributions from chromosome IX loci is not due to the aneuploidy (Fig. 3 and Supplementary Figure 3), since all aneuploidies were excluded in does have a large effect on many of the phenotypes.

The Methods section has been altered and now reads:

“The theoretical maximum amount of POLs from our experimental design was 7396 (86×86); however, one F12 haploid strain (MAT α , number 45) was contaminated prior to mating and all 86 hybrids spawning from this cross were therefore discarded ($86 \text{ MATa} \times 85 \text{ MAT}\alpha = 7310$ were retained). Furthermore, 8 F12 haploids were identified as having chr. IX aneuploidy (see Genotype construction below), the hybrids spawning from these haploids were included in the phenotyping in order to investigate the aneuploidy's effect on the phenotype. They were, however, excluded in all downstream analysis since they could interfere with the QTL mapping and they have a large fraction of missing genotypes on chr. IX. We do find a possible effect of the chr. IX aneuploidy mainly on the mean growth phenotype (see Supplementary Fig. 1a, bottom panel).”

“Chr. IX aneuploidy was identified based on higher sequencing coverage and higher fraction of heterozygous polymorphic sites compared to the genome as described in Cubillos et al. (2013)¹⁷. The following eight haploid F12 parents carried the aneuploidy: MAT α 41, 53, 67 and MAT α 206, 222, 223, 253, 258. Contaminated diploid hybrids and hybrids with chr. IX aneuploidies were excluded. Phased genomes of the 6642 diploid hybrid offspring (81 MAT α x 82 MAT α) retained for the genetic analysis was constructed in silico using custom R code.”

Minor comments

1. "More QTLs (69%) were shared between maximum doubling time and mean growth than expected from their low general correlation (mean r : 0.27, Fig. 1d)." How many are expected from the correlation? How is such an expectation calculated?

Author response: Our phrasing was misleading here. We have amended this and the section (page 4, lines 166-167) now reads:

“A surprisingly large number of QTLs (69%) were shared between growth rate and mean growth, given that the overall correlation between these growth variables was low (mean r : 0.27, Fig. 1d).”

2. "Raw population growth curves were slightly smoothed to remove noise"
How was this done?

3. "Poor quality curves (0.3%) were rejected following manual inspection."
What constituted poor quality?

Author response: The procedure for normalization of growth curves and quality control is complex. The procedure is extensively described and discussed in the recent Method paper referenced. We have, however, expanded the text section providing an abbreviated description of the method. The section (page 7, lines 334-344) now reads:

“Calibrated pixel intensities were transformed into population size measures by reference to cell counts obtained by optical density measurements on diluted samples. Raw population growth curves were slightly smoothed using a median (size = 5) and a Gaussian (width σ = 1.5) filter to remove noise. Poor quality curves (1%, resulting from e.g. positions lacking colonies) were rejected following manual inspection²⁰. Retained population growth curves were broken down into two growth phenotypes: i) growth rate, extracted using linear regression from the steepest slope of the population’s exponential phase, and ii) mean growth, extracted as the area under the curve relative to its starting point but excluding the three first time points. To counter spatial bias on each 1536 plate, the two growth phenotypes were normalized to the internal controls using the Scan-o-matic principle²⁰. The final phenotypes used were the average phenotype across all replicates. Detailed protocols are available for the entire phenotype acquisition²⁰”

4. Fig 1 legend: "gorwth" typo

Author response: Resolved

Reviewer #2

In this MS an analysis is presented of a large reciprocal F1 cross between two independent sets of 86 haploid yeast strains, ie 7396 potential F1 offspring. The $2 \times 86 = 172$ parents were F12 haploid recombinant inbred lines from a two-parent cross. Thus in effect the chromosomes are all mosaics of a two-haplotype system. Growth traits were measured in replicates in these offspring under different environments. Genetic analysis to partition heritability and map QTLs was performed.

The idea of making what is in effect a large diallele cross is not new - see for example the RIX paper <http://www.ncbi.nlm.nih.gov/pubmed/26669442> for a theoretical analysis - but so far as I know this is the first study to actually collect phenotypic data and to analyse it. It would therefore be of interest to the statistical genetics readership of Nature Communications.

Judging by the large heritabilities obtained, the phenotyping seems to have performed with great accuracy. The results as presented are focused on statistical genetics questions - estimates of heritability, heterosis etc. Although many QTLs are mapped, it's unclear if the functions of any novel genes have been confirmed.

Author response: We appreciate the reviewer's suggestion and have looked deeper into the RIX related literature. Although, we assume that the wrong link was pasted (as it leads to a paper with a quite different focus), we found the original paper that introduce the RIX approach and subsequent papers. As the reviewer pointed out, the few RIX studies published so far have remained mainly theoretical, but do provide an important backdrop to this paper. We now discuss and reference the RIX approach in the introduction (page 1, lines 53-56) which now reads:

“Inspired by previous thinking and theoretical work on recombinant inbred intercusses in other model organisms¹³⁻¹⁵, we here introduce a powerful and cost-effective framework for tracking the covariation through genome and phenome that allows accurate estimates of dominance and epistasis in diploid models”

We want to emphasize that we do not set out to reveal any novel biology but to validate the power and accuracy of the experimental approach for mapping QTLs and investigating the nature of genetic interactions. The examples we use for validation (the GAL3 gene and DAL cluster) were in fact validated previously.

1. The paper is extremely brief, which makes parts of it very hard to follow. I think the page limits for Nature Communications Articles may be partially responsible but there is an over-long discussion at the cost of very opaque results (I found the discussion of heterosis in lines 187-203 particularly difficult). Similarly it was only after carefully reading the supplemental material that it became clear that the experimental design involves 86 haploids from each mating type - I initially thought that a full diallele cross of 172 lines was being analysed.

Author response: We have expanded the entire text to make the manuscript clearer, giving special emphasis to the points brought up here. For heterosis see our response to comment nr. **5** and for the diallele cross design see our response to comment nr. **2**.

2. Although the experimental design is presented as being generally applicable outside of the yeast genetics community, many of the details would be unclear to non-specialists (eg the constraints due to mating type). For example, if one were to repeat this design in a monoecious species such as Arabidopsis, there would be no need to distinguish the two sets. It's also worth pointing out the fact that in a diallele of diploid RILs (eg mice, Arabidopsis) the F1 genotypes are still trivial to infer from the parents.

Author response: To accommodate the reviewer's concerns regarding the opacity of the experimental design descriptions (comment 1 and 2) we have modified the first paragraph of the results (page 2, lines 66-73). It now reads:

“To accurately decompose diploid trait variation, we first isolated and sequenced the full genomes of 86 MAT α and 86 MAT α haploid yeast strains. These haploids were randomly drawn from a 12th generation two-parent intercross pool, constructed using highly diverged (0.53% nucleotide difference) wild strains, here termed North American (NA) and West African (WA). Only two alleles segregate at each polymorphic site, with on average equal representation in the pool¹⁶. The sequenced haploids of opposite mating types were systematically crossed in all possible pairwise combinations to generate 7396 genetically distinct diploid hybrids, retaining 6642 POLs used for all downstream analysis (Fig. 1a, Supplementary Fig. 1a, Methods).”

We are now explicit with that a full diallel cross was not constructed. We also expand the discussion (page 5, lines 249-254) and methods (page 6, lines 290-296) to further clarify this aspect:

“Beyond the removal of the sex-switch (*HO* gene) and introduction of sex-specific auxotrophic markers, POLs impose no requirements on the yeast genotypes used; the design is lineage agnostic. However, removal of the yeast sex-switch renders the cross directional and prevents the construction of a full diallel cross, something that is otherwise possible in for example monoecious plants where individuals express both sexes.”

“Haploid genotypes were selected to allow systematic crossing: MAT α , *ura3::KanMX*, *ho::HygMX* and MAT α ; *ura3::KanMX*; *ho::HygMX*; *lys2::URA3*. Haploids of different mating types were robotically mated on rich medium (1% yeast extract, 2% peptone, 2% glucose, 2% agar) in all pairwise combinations combining their complementary *LYS* and *URA* auxotrophies using a RoToR HDA robot (Singer Ltd, UK). Haploid cells of the same mating type do not mate and this feature prevents the construction of a full diallel cross (e.g. MAT α /MAT α and MAT α /MAT α diploid hybrids cannot be constructed).”

Regarding the F1 genotypes being trivial to infer in diploid RILs, we now make it clear that this methodology can be used with models other than yeast and that this represents a strength of the method.

3. The introduction of the (apparently) new terminology POL (phased outbred line) seems unnecessary and is puzzling as it over-emphasizes the phased nature of the genotypes. Although the genomes are trivially phaseable with respect to the parents, no use is made of this phasing in the MS. For example, it would be straightforward to test for parent-of-origin effects by exploring asymmetries in the phenotypes of lines carrying the phased genotype XY at a given locus vs those carrying genotype YX. Perhaps the authors did this but did not find anything? It would be interesting to know. Similarly, the long range phasing in this population means it would be possible to explore epistasis where there is a difference between a pair of loci with phased genotypes (AB - AB) compared to (AB- BA). One could presumably

partition the epistatic component of heritability into these sub-components (the methodology described in the supplemental note suggests this would be possible).

Author response: We recognize the reviewer's point about the choice of the name Phased Outbred Lines, however, we deem the new terminology warranted. We considered using the existing RIX, but decided against it since our strains are not strictly recombinant inbred, and we do not apply a full diallel cross. Our lines stand out compared to the RIX approach and the immortalized F2 population studies with respect to number of recombination events and genotype density throughout the genome. Further, we can infer near complete genotypes, since we have no remaining segregating heterozygosity, and the mitochondrial genome is inherited from only one of the two parents. Even though we do not make extensive use of it in the current manuscript, the phased genome is an important and noteworthy characteristic of our hybrid population with potential applications that will be explored in the future. Taking all of the above into account, we would like to keep the POL name, but are open to changing it if the Editor finds it to be necessary. We do agree with the reviewer that we put a too great emphasis on the phased nature of the genomes, and have removed "phased outbred lines" from the title which now reads:

"Powerful decomposition of complex traits in a diploid model"

We appreciate the suggestion made on the analysis for parent-of-origin effects. Indeed, this has been shown to be an important contributor to complex trait variation in other mapping populations (e.g. in outbred mice: <http://www.ncbi.nlm.nih.gov/pubmed/24439386>). We explored the referee's idea in our current dataset for all the 145 QTLs as well as for 145 randomly selected regions and detect some potential cases of parent of origin in both sample set. This was done by dividing the heterozygous genotypes as either NW or WN, with the allele contributed by the *MAT a* strain first (NW) and the allele contributed by the *MAT α* second (NW). We found significant differences (Students t-test, Bonferroni corrected $q < 0.01$) between the two heterozygous states for around 60% of QTLs, for the same amount of randomly selected markers, we found that around 50% differed significantly between the two heterozygous states. Comparing the effect size difference between the two heterozygous states and the two homozygous states, we find that the effect size of the two heterozygous states are on average only 20% of that of the two homozygous states. However, this type of analysis suffers from population structure and varying levels of genome relatedness. Applying parent-of-origin analysis genome wide is therefore a major methodological undertaking that will require extensive and lengthy experimental validation. This is beyond the scope of the current paper.

Moreover, parent-of-origin effects might play a much smaller role in yeast than in higher organisms, given that its major underlying mechanism is genomic imprinting which has never been documented in budding yeast. *S. cerevisiae* yeasts spend most of their life cycle as diploids *MATa/α* (without sex distinction among diploids) with approximately 90% of wild strains that are isolated as *MATa/α* diploids (including the founder strains North American and West African used in our study). We artificially derived stable *MATa* and *MATα* haploid lines, and therefore we can apply the parent-of-origin analysis suggested by the reviewer. However, this mating type configuration is only superficially reminiscent of the mother and father sex determination since it is exclusively expressed in haploid cells, which are believed to be a very short stage of the natural *S. cerevisiae* life cycle. Rather than the classic parent-of-origin effect, our hybrid POLs might indeed be useful to detect *MATa* and *MATα* specific phenotypic effects. A small subset of mating specific gene expression differences and phenotypes have been

documented between *MATa* and *MATα* haploids (Galitski *et al.* 1999 <http://www.ncbi.nlm.nih.gov/pubmed/10398601>). However, these differences are likely to be transient and not transmitted across generations in this single cell organism and we expect to lose this effect during the hybrid construction before the phenotyping. Since we did not set out to investigate this mating specific effect, the crossing step was followed by multiple rounds of growth in diploid selective media to ensure clean diploid populations.

In Figure 2 heritability is partitioned into additive, dominance, and 2-way epistatic components. 3-way components are plotted separately. Why was this? In the supplemental note simulations suggest that incorporating 3-way interactions distorts the estimation of the other variance components. Was this the reason? By eye it looks like the 3-way epistatic variance must be confounded with some of the other components since the total variance when 3-way epistasis is included appear to exceed 100%. In the upper panel of Figure 2, it would be helpful to add lines on the bars indicating the total variance explained by mapped QTLs.

Author response: We separate the variance decomposition plot (Fig. 2) in order for it to comply with the analysis made. As third order interactions were estimated separately, we consider separating the plot to be the most truthful way of displaying the data. We have attempted to make the analysis of the variance decomposition more clear by expanding the relevant section in the Methods (pages 7-8, lines 350-360), which now reads:

“Phenotype variance partitioning: We estimated additive relatedness from genotypes. We derived formulae for efficient computation of the covariance due to dominance, pairwise and third order interaction effects (Supplementary Note 1). We fitted the model using restricted maximum likelihood, as in Yang *et al.* (2011)⁴⁹. The variance decomposition and its associated standard errors were found to be accurate and close to unbiased in simulations when fitting additive, dominance, and pairwise interaction components (Supplementary Note 1). However, when adding a component for third order interactions, the overall variance decomposition became biased, even though the estimates of the third order component did not. We believe this may be the result of non-convexity in the optimization problem, as evidenced by bimodality in the distribution of estimates of pairwise interaction variance in simulations including the third order component. We therefore report estimates of the variance from third order interactions separately from the decomposition into additive, dominance, and pairwise interaction components.”

We would prefer not adding the variance explained by QTLs to Figure 2 as the two analyses are different and do not represent the same thing. The QTL calling is partitioned into additive and nonadditive, this means that additive QTLs only explain a fraction of the additive variance, while nonadditive QTLs only explain a fraction of the nonadditive variance. The variance decomposition on the other hand looks at the entire phenotypic variation. To put the variance explained by QTLs and the variance decomposition on the same scale would not truly reflect what we are measuring. However, we agree that the variance explained by the QTLs is interesting and worth a place in the manuscript. Therefore, we added a supplementary figure (Supplementary Figure 5) to show this.

4. As the authors point out, but in a somewhat cryptic manner, and as other have pointed out (eg in the RIX paper cited above), the problem with the POL/RIX design is that the number of independent genomes is given by the number of parents (in this case 172). This means that all pairs of F1 individuals that share a common parent have much more allele sharing than two individuals that don't. Consequently the entries of the genetic relationship matrix of the population have a bimodal distribution (trimodal if one includes the main diagonal - it would be interesting to see it as a supplemental figure). The population

resembles a large population of halfsibs in which every male mates with every female. It is implicitly claimed that the mixed model used to do the QTL analysis adequately controls for the background relatedness but I would like to see some evidence. Some QQ-plots for a few genome scans would help. Similarly, does the design introduce any biases in the estimates of heritability? Is it certain that epistasis is estimated correctly, given the long-range correlations in the genotypes? I guess the simulations in the supplemental note address this, but this should be discussed in the main text

Author response: The referee is correct about the bimodal distribution of the genetic relationship, which can be found in our recent paper [Märtens *et al.*] that explores phenotype prediction using the dataset generated here. We have added the following to the main text (page 2, lines 82-85):

“Hybrid pairs sharing one haploid parent will be genetically more similar than two POLs that do not share a parent (expected fraction of loci with identical genotypes = 0.5 and 0.375 respectively), resulting in a bimodal distribution of the genetic relationship matrix entries¹⁹.”

Furthermore, we welcome the suggestion to add a proof of principle for the linear mixed model accounting for population structure and have included a new supplementary figure (Supplementary Figure 8) with QQ-plots for the two QTLs we use as examples in figure 4. The method section (page 8, lines 377-384) now reads:

“We accounted for population structure in the LIMIX analysis by using the genetic relationship matrix defined by where X is a centered and standardized genotype matrix, and the normalizing constant c is the average diagonal value of XX^T . This is in contrast to the mapping in R/qlt where we instead modified the phenotype used, as stated at the beginning of this section. QQ-plots (Supplementary Fig. 8) confirm that the linear mixed models appropriately account for population structure: apart from the locus with the strongest effect (*DAL* and *GAL* loci, in allantoin and galactose respectively), the distribution of the rest of p -values follows the expected uniform distribution under the null.”

In response to potential bias in the variance decomposition, this approach to estimate epistatic variance components has been used before in haploid yeast populations exhibiting even greater long-range linkage disequilibrium (Young and Durbin, Ref 12; Bloom *et al.* Ref 10). The variance decomposition of a trait always relies to some degree on the linkage present in the population of genomes analysed, even for additive effects. Standard additive mixed models are able to deal with this. This is true also for epistasis. Consider two loci that may interact with each other, but are in perfect linkage equilibrium. They will exhibit only additive statistical variance as the variation in the interacting components is never observed in the population. The mixed model approach will estimate the variance from interactions, which is a function of their linkage. Furthermore, the simulations show that epistatic variance is estimated accurately for evenly spaced loci, some of which will be in linkage. In addition to the extended method and results presented in the **Supplementary Note 1**, the result section “**Near complete variance decomposition of diploid traits**” (page 3, lines 106-135) and related method section “**Phenotype variance partitioning**” (pages 7-8, lines 350-360) has been extensively edited and expanded.

5. As noted above, the discussion of heterosis is hard to follow. I wonder if would be better to simplify it and not subdivide it into over and under-dominance. Some way of displaying the variance attributable to heterosis in Figure 2 would be helpful. In this study, the phenotypic value of a given parent is estimated by taking the mean of all F1s with that parent. This definition is sensible from the point of view of

establishing genome-wide levels of heterosis, but if one is investigating heterosis at a particular QTL then perhaps it is better to estimate the expected phenotypic value of AA genotypes vs BB genotypes (where A is the allele carried by the NA strain and B that by the WA strain), from the means of all F1s that are AA vs BB. Would this definition have resulted in substantially different conclusions regarding heterosis?

Author response: We agree that the section on heterosis and dominance was probably too concise and difficult to follow in the original version. We have made extensive changes to this section titled “Explaining heterosis by intralocus interactions” (pages 4-5, lines 177-230)

It is correct that we estimate a given parents phenotype by taking the average of all its spawned hybrids, and that we later use these values to estimate heterosis in the hybrids. Heterosis is the significant phenotype deviation of a diploid hybrid genome relative its two parental genomes. Potentially, the phenotypic variance that is contained in such significant deviations could be estimated, but this gives a somewhat misleading picture because variation among significant BPH cases may be low, even if their mean deviation from the parents is high and the heterosis consequently large. Moreover, these estimates have no immediate connection to the variance decomposition in Fig 2 and would complicate its interpretation, without providing much additional relevant information. The alternative view on heterosis that is contemplated by the reviewer, considering QTL contributions to heterosis on the basis of the mean phenotype of NA homozygotes, WA homozygotes and NA/WA heterozygotes can be implemented in two different ways, and we are not sure which the reviewer envisions. However, they both come with serious caveats attached. Means for the three genotype groups may be compared directly. We now try to make it clearer why this is a less promising approach than the one taken by us (page 5, lines 195-200):

“However, calling such contributions is challenging because multiple effects often act in parallel. In particular, overdominance may be modified by epistasis such that it only manifests in a minority of genetic backgrounds³¹. Thus, a QTL may not be overdominant in the average genetic background, but could nevertheless account for best parent heterosis in some lineages. Comparing the mean phenotypes for heterozygous and homozygous genotypes is therefore a blunt tool for detecting overdominant contributions to best parent heterosis.”

We also report the outcome of such an alternative approach (page 5, lines 215-218):

“The dominance/overdominance contributions of QTLs to best parent heterotic POLs were often notably different from their contributions to the population as a whole (Fig 4b). Only two of the 14 QTLs for which we detected overdominance in the best parent heterotic POLs had, on average, a significantly superior heterozygote state when the entire POL population was considered (Student’s t-test, $p < 0.01$).”

The other potential implementation of the contemplated alternative view is to estimate what fraction of NA/NA genotypes that deviates in a positive direction from the NA/NA mean. These fractions could then potentially be compared across the three genotype classes, and significance established such that e.g. disproportionately common positive deviations of NA/WA individuals from the NA/WA mean is taken as an indication of overdominance potentiated by epistasis. However, this implementation involves no connection to or assumptions of heterosis.

We have therefore clarified and re-written this paragraph, but do not find it motivated to implement the alternative analyses contemplated by the reviewer.

6. Was the set of SNPs tested in the GWAS - 56k in the R/qtl analysis, followed by only 10k in LIMIX after pruning - the total number segregating in the population? What was the advantage of using 56k SNPs in the first stage is the pruning removed SNPs perfectly tagged (in the LD sense) with others? Were there more SNPs detected during sequencing? If so, what was done with them? the numbers of SNPs tested should be mentioned in the main text.

Author response: The reviewer is correct in that we used different amount of SNPs for the mapping in the R/qtl and the linear mix models (LIMIX). The purpose of the removal of SNPs in the LIMIX was increased computational speed (which is mentioned in the Methods), this was not necessary for the R/qtl and was therefore not applied. Subsequently the SNPs used for the R/qtl was the full set of SNPs. We have amended the Methods as follows (page 8, lines 362-366):

“QTL mapping: QTL calling was made using the scanone function with the marker regression method in R/qtl⁵⁰ with estimated diploid parent phenotypes (additive genetic background contribution to traits) and POL deviations from the estimated diploid parents values (variation not explained by additive effects of parental background) respectively using the full set of 52,466 markers (including redundant markers).”

And we have added the number of SNPs used for the R/qtl in the main text (page 3, lines 143-145) :

“We mapped QTLs using 52,466 markers, the inferred parent phenotypes (for additive effect of genetic background) and the hybrids deviations from the average of the inferred parental phenotypes (for nonadditive effects; Methods).”

7. More details of the R/qtl mixed model used for QTL mapping should be given. What was the definition of the kinship matrix? Was this the same as that used in LIMIX? The choice of the LOD drop threshold used to define confidence intervals should be justified.

Author response: We have added information in the Methods section to further explain what the reviewer points out here. We have added the definition of the kinship matrix used in the mapping with LIMIX (page 8, lines 377-380) and tried to make it more clear how we take population structure into account in the R/qtl analysis, where no kinship matrix was used (page 8, lines 362-369). The relevant section in methods now reads:

QTL mapping: QTL calling was made using the scanone function with the marker regression method in R/qtl⁵⁰ with estimated diploid parent phenotypes (additive genetic background contribution to traits) and POL deviations from the estimated diploid parents values (variation not explained by additive effects of parental background) respectively using the full set of 52,466 markers (including redundant markers). Diploid parental phenotypes were estimated as the median of all hybrids descended from that parent. Using the deviations from expected midparent phenotype for the POLs has the additional critical benefit of effectively accounting for population structure by removing the additive effect of the more similar genetic composition due to shared parents.

“We accounted for population structure in the LIMIX analysis by using the genetic relationship matrix defined by where X is a centered and standardized genotype matrix, and the normalizing constant c is the average diagonal value of XX^T . This is in contrast to the mapping in R/qtl where we instead modified the phenotype used, as stated at the beginning of this section.”

We agree with the reviewer that we should justify our choice of LOD support interval. We choose a 1.8 LOD drop threshold since that is the one preferred for intercrosses by Karl Broman in his guide to QTL mapping. We have added this in the Methods (page 8, lines 369-372):

“Significance thresholds were given by permutations (x1000), 1.8-LOD support intervals were calculated for each QTL using the lodint function in R/qtl, this corresponds to the LOD support interval stated as the preferred one for intercrosses in A Guide to QTL Mapping by Broman *et al.*⁵⁰”

Reviewer #1 (Remarks to the Author):

The authors have addressed my concerns.

Reviewer #2 (Remarks to the Author):

The authors have answered most of my queries satisfactorily. It's a nice paper. I don't have any major points to make. A few minor queries that should be clarified:

1. Abstract: the phrase "overdominant and pervasive pleiotropy" is unclear.
2. Line 161 This is not my understanding of pleiotropy, which is where a single locus affects multiple traits. The authors are using it to mean a single locus affects the same trait in different environments.
3. line 184 the accepted definition of heterosis is surely different from that implied here, which appears to simply be non-additivity.

Also I don't insist but it would be nice in the discussion if some mention of the experimental design's applicability to parent of origin effects was made.

It's a nice paper.

Response to reviewer and editorial comments

Reviewer comment

Author response

Revised manuscript text

Reviewer #1 (Remarks to the Author):

The authors have addressed my concerns.

Reviewer #2 (Remarks to the Author):

The authors have answered most of my queries satisfactorily. It's a nice paper. I don't have any major points to make. A few minor queries that should be clarified:

1. Abstract: the phrase "overdominant and pervasive pleiotropy" is unclear.

Author response: We agree with the reviewers that this phrase was unclear, pervasive pleiotropy refers to the QTLs and does not have anything to do with the dominance and overdominance. The text has therefore been revised to read:

“We map quantitative trait loci (QTLs) and find nonadditive QTLs to outnumber (3:1) additive loci, dominant contributions to heterosis to outnumber overdominant, and extensive pleiotropy.”

2. Line 161 This is not my understanding of pleiotropy, which is where a single locus affects multiple traits. The authors are using it to mean a single locus affects the same trait in different environments.

Author response: Pleiotropy takes slightly different meanings in different fields, largely due to that phenotypes on very different levels are considered. In this paper, the traits we consider are microbial population growth in different environments. The established standard in microbial genetics is to refer to growth in different environments as different traits, at least when these environments affect growth differently. Effects of single gene or mutation across multiple such environments are thus typically referred to as pleiotropy. See for example Bloom *et al.* 2013, nature.com/nature/journal/v494/n7436/full/nature11867.html, as a leading paper in the field. We consider it most prudent to continue adhering to this standard.

3. line 184 the accepted definition of heterosis is surely different from that implied here, which appears to simply be non-additivity.

Author response: Heterosis is, and has been defined differently, by different persons and in different fields of genetics. The broadest definition of heterosis is the one we refer to in this sentence: non-additivity relative the midparent expectation. This is often referred to as midparent heterosis. Midparent heterosis is then further resolved into positive and negative mid-parent heterosis, which have the subcategories best and worst parent heterosis (where the latter two is defined not by comparison to parental averages but to the best and worst of the parents, respectively). The oldest definition of heterosis equated heterosis with best parent heterosis, which is what the reviewer is referring to. To avoid potential misunderstandings in this respect, we have revised the text to read:

“Offspring where the inferred parents differed significantly were eligible for discordance analysis, the majority of the hybrids (89% to 95%, depending on threshold) that could be unambiguously called (Methods and Supplementary Fig. 7a) deviated significantly from the midparent expectation and were thus midparent heterotic.”

Also I don't insist but it would be nice in the discussion if some mention of the experimental design's applicability to parent of origin effects was made.

Author response: We agree that this would make a nice addition and have added a sentence in the Discussion on parent of origin:

“Furthermore, genome phasing information in POLs derived from higher organisms is ideal for investigating parent-of-origin contributions to complex trait variation⁴⁸.”

Editorial comments

Please revise the title to be free of adjectives

You suggest to remove the adjective “powerful” from the title of the manuscript. While we appreciate and understand the reasoning behind not using adjectives in the title, as they can be quite subjective, we are of the opinion that in our case an adjective is appropriate. Our manuscript puts a large emphasis on the POL resource as a way to make the most out of studying variation in diploid populations. In fact, the increase in power that follows from the POL approach is the single most important advancement we report here. To emphasize that, we chose to begin the title with “Powerful”. The increase in power is not a subjective judgement open to contention: it is well supported by numbers throughout the abstract and the text. We also consider removing “Powerful” to the title rather unappealing and incapable of attracting attention. Finally, we do note that adjectives that are far more contentious and with much less empirical support, e.g. “ultrafast”, “efficient” and “superior” are quite common in recent Nature Communications articles (e.g. PMID: 27558837, 27539942, 27516157, 27515900, 27515779, 27514992).

Please provide postal codes for all AU affiliations

Postal codes have been provided

Abstract and text - Please spell out the exact yeast species used here.

“Yeast” has been replaced by *Saccharomyces cerevisiae* in both instances.

Methods page 7, line 317 - For transparency and reagent availability, please consider sharing the custom R code here in a public depository and indicate its accessibility in the Methods text.

Associated R code has been deposited in a public repository (<https://github.com/j-hallin/y10k>), which has been indicated in the Methods.

Methods page 8, line 362 - please use font that is consistent with the rest of the text. If these names are unique, consider using "" instead.

The font of the functions within the R/qtl package was changed knowingly to differentiate between regular text and actual functions in the R/qtl package. However, we have changed it to be surrounded by quotation marks instead

Please make sure that this and all other equations are provided in an editable format (e.g. equation editor in MS Word).

All equations are in Words equation format

Supplementary Information Order:

A Supplementary Information file accompanies the submission in the specified order. All references has been moved out of Supplementary Table 1 and Supplementary Note 1 to together occupy the final pages of the Supplementary Information file.

Supp tables 2 and 3 are large and may be better off as supplementary data. Please convert.

Supplementary table 2 and 3 have been changed to supplementary data 4 and 5 respectively

If possible, please convert these CVS format to .txt or .xls format as the latter two are more compatible with our system.

All .csv files have been converted into tab separated .txt files

Figure legend should be included as a part of main text. Figures themselves should be uploaded to our system individually. Please see the decision letter for more details on figure preparation.

All figures have been transformed into .ai files with text in a separate layer. The figure legends are located in the main text